# Serum Biomarkers for Connective Tissue and Basement Membrane Remodeling Are Associated with Vertebral Endplate Bone Marrow Lesions as Seen on MRI (Modic Changes)

**DOI:** 10.3390/ijms21113791

**Published:** 2020-05-27

**Authors:** Stefan Dudli, Alexander Ballatori, Anne-Christine Bay-Jensen, Zachary L. McCormick, Conor W. O’Neill, Sibel Demir-Deviren, Roland Krug, Irina Heggli, Astrid Juengel, Jaro Karppinen, Florian Brunner, Mazda Farshad, Oliver Distler, Jeffrey C. Lotz, Aaron J. Fields

**Affiliations:** 1Center of Experimental Rheumatology, University of Zurich, Balgrist Campus, 8008 Zurich, Switzerland; irina.heggli@usz.ch (I.H.); astrid.juengel@usz.ch (A.J.); oliver.distler@usz.ch (O.D.); 2Department of Orthopaedic Surgery, University of California San Francisco, San Francisco, CA 94142, USA; alexander.ballatori@ucsf.edu (A.B.); zachary.mccormick@ucsf.edu (Z.L.M.); conor.oneill@ucsf.edu (C.W.O.); sibel.demir@ucsf.edu (S.D.-D.); roland.krug@ucsf.edu (R.K.); jeffrey.lotz@ucsf.edu (J.C.L.); Aaron.Fields@ucsf.edu (A.J.F.); 3Immuno-Science, Nordic Bioscience, Biomarkers and Research, 2730 Herlev, Denmark; acbj@nordicbio.com; 4Medical Research Center Oulu, Oulu University Hospital and University of Oulu, 90220 Oulu, Finland; jaro.karppinen@ttl.fi; 5Center for Life Course Health Research, University of Oulu, 90220 Oulu, Finland; 6Finnish Institute of Occupational Health, 90220 Oulu, Finland; 7Department of Physical Medicine and Rheumatology, Balgrist University Hospital, 8008 Zurich, Switzerland; florian.brunner@balgrist.ch; 8Department of Orthopeadic Surgery, Balgrist University Hospital, 8008 Zurich, Switzerland; mazda.farshad@balgrist.ch

**Keywords:** Modic change, low back pain, biomarker, connective tissue, basement membrane, bone marrow, disc degeneration

## Abstract

Vertebral endplate bone marrow lesions, visualized on magnetic resonance imaging (MRI) as Modic changes (MC), are associated with chronic low back pain (cLBP). Since guidelines recommend against routine spinal MRI for cLBP in primary care, MC may be underdiagnosed. Serum biomarkers for MC would allow early diagnosis, inform clinical care decisions, and supplement treatment monitoring. We aimed to discover biomarkers in the blood serum that correlate with MC pathophysiological processes. For this single-site cross-sectional study, we recruited 54 subjects with 38 cLBP patients and 16 volunteers without a history of LBP. All subjects completed an Oswestry Disability Index (ODI) questionnaire and 10-cm Visual Analog Score (VAS) for LBP (VASback) and leg pain. Lumbar T1-weighted and fat-saturated T2-weighted MRI were acquired at 3T and used for MC classification in each endplate. Blood serum was collected on the day of MRI. Biomarkers related to disc resorption and bone marrow fibrosis were analyzed with enzyme-linked immune-absorbent assays. The concentration of biomarkers between no MC and any type of MC (AnyMC), MC1, and MC2 were compared. The Area Under the Curve (AUC) of the Receiver Operating Characteristics were calculated for each biomarker and for bivariable biomarker models. We found that biomarkers related to type III and type IV collagen degradation and formation tended to correlate with the presence of MC (*p* = 0.060–0.088). The bivariable model with the highest AUC was PRO-C3 + C4M and had a moderate diagnostic value for AnyMC in cLBP patients (AUC = 0.73, specificity = 78.9%, sensitivity = 73.7%). In conclusion, serum biomarkers related to the formation and degradation of type III and type IV collagen, which are key molecules in bone marrow fibrosis, correlated with MC presence. Bone marrow fibrosis may be an important pathophysiological process in MC that should be targeted in larger biomarker and treatment studies.

## 1. Introduction

Modic changes (MCs) are MRI signal intensity features within vertebrae adjacent to degenerated intervertebral discs [1,2,3]. There are three interconvertible types that are defined according to their appearance on T1-weighted and T2-weighted MR images [2]. In a systematic review, the prevalence of MC in patients with chronic low back pain (cLBP) was 43% (in contrast to 6% in the non-clinical population) [4] but varied greatly with geographic location. Generally, prevalence is lower in Asia (China and India: 10.9–22.4%) [5,6,7] and Africa (Nigeria: 22.8%) [8]. Prevalence increases with age and is most frequent at lower lumbar levels, i.e., L4/L5 and L5/S1 [2,9]. Endplate defects predict MC and suggest biomechanical aspects as important etiological parameters of MC [3,10].

MC have a high specificity for discography-concordant cLBP [11]. In contrast to disc herniation, where nerve compression causes back and leg pain, MC pain is believed to arise from pathologic innervation in the endplate and vertebra [11,12,13]. From a clinical population-based study, including 412 Danes, Kjaer et al. concluded that “MC in combination with disc degeneration is an entity on its own, which is different from disc degeneration without MC” [14]. LBP patients with MC report a greater frequency and duration of LBP episodes, seek care more often and have a higher risk for a poor outcome [4,15,16,17,18]. The larger the MC lesions, the lower the chance for resolution, and the higher the positive predictive value for pain in discography [9,19]. Therefore, early detection of MC may be important for informing clinical decisions and treatment monitoring [20,21,22]. However, MC may be underdiagnosed in primary care because clinical guidelines recommend against routine spine MRI for “unspecific” cLBP. Serum biomarkers are a fast and inexpensive way to detect pathological conditions. A robust MC biomarker should reflect the pathophysiological processes (face validity).

MC have been described as inflammatory changes with fibrosis, increased vascular supply, and high bone turnover [2,10,23,24]. Discs adjacent to MCs degenerate at an increased rate and secrete higher levels of pro-inflammatory cytokines and aggrecanases [10,24,25,26,27,28]. No biomarkers for MC have yet been identified. Yet, a small French study reported elevated high-sensitivity C-reactive protein (hsCRP) in patients with Modic type 1 changes (MC1) [29]. HsCRP is a non-specific marker for inflammation and hence not a robust biomarker. A recent study on 80 subjects investigated the concentrations of 46 serum cytokines between MC patients and healthy volunteers [30]. Interestingly, 16 cytokines related to inflammation, bone turnover, angiogenesis, and vascular injury were downregulated in MC. In a case-control study, with 13 MC1 patients and 21 control cLBP patients without MC, Boisson et al. found no difference in serum markers of inflammation (interleukin (IL)-1β, IL-6, IL-8, tumor necrosis factor alpha (TNF-α)) [31]. This stands in contrast to the current ideas about MC pathophysiology. Therefore, blood serum cytokines may have little face validity for MC. Blood biomarkers for oxidative stress (total thiols, advanced oxidation protein products, and carbonyl groups) in MC1 patients are inconclusive [31,32] and degradation markers for type II collagen (Coll2-1, Coll2-1NO_2_), important disc collagen, were not elevated in MC1 patients [31]. Taken together, these studies do not yet validate the concept of ‘active discopathy’, defined as an accelerated disc degeneration with low grade local and systemic inflammation [33]. However, no study has yet investigated biomarkers related to bone marrow fibrosis, which is a histological hallmark of MC.

We aimed to detect biomarkers reflecting connective tissue and basement membrane turnover for MC that have high face validity. These biomarkers may relate to the rapid resorption of the adjacent disc and the fibrotic changes of the bone marrow. The disc matrix consists mainly of type I and II collagen, as well as aggrecan. Type III collagen is present only in traces in the pericellular environment of disc cells and type IV collagen is absent [34,35]. Therefore, resorption of the disc generates neo-epitopes of type I and II collagen and aggrecan. In myelofibrosis (a fibrotic condition of the bone marrow different from MC), type I and III collagen are the major deposited matrix constituents [36]. Neovascularization of fibrotic bone marrow increases the spatial abundance of type IV collagen because type IV collagen is the main constituent of the basement membrane of vascular sinusoids [37,38]. Serum procollagen type III and IV are known to be increased in myelofibrosis [39]. Therefore, biomarkers for type III and IV collagen, but not type I and II collagen and aggrecan, may relate to bone marrow fibrosis. In this current study, we quantified the association of nine matrix biomarkers related to disc resorption and bone marrow fibrosis (neo-epitopes of type I to IV pro-collagen (PRO-C1, PRO-C2, PRO-C3, PRO-C4), type I to IV collagen degradation neo-eiptopes (C1M, C2M, C3M, C4M, aggrecan degradation neo-epitope (huARGS)) to lumbar MC. Since a previous study reported no increase in type II collagen degradation markers [31], we hypothesized that biomarkers related to bone marrow fibrosis are increased in MC.

We observed that serum biomarkers related to the formation and degradation of type III (PRO-C3, C3M) and type IV collagen (PRO-C4, C4M), correlated with MC presence. This underscores that bone marrow fibrosis is an important pathophysiological process in MC, and suggests that it should be a focus in larger biomarker and treatment studies.

## 2. Results

### 2.1. Study Subjects

There were no significant differences in sex, age, and body mass index (BMI) between cLBP patients and control subjects or between subjects with any type of MC (AnyMC) and without MC (Table 1). Oswestry Disability Index (ODI), visual analogue score (VAS) for back pain (VASback), and VAS for leg pain (VASleg) were significantly higher in the patient group than in the control group but not different between LBP patients with AnyMC and without MC. The occurrence of MC was higher in the patient group (50.0% vs. 12.5%, *p* = 0.01), mainly because of increased presence of MC1 in the patient group (26.3% vs. 0%, *p* = 0.03. MC2 (28.9% vs. 12.5%, *p* = 0.30) and MC3 (5.3% vs. 0%, *p* = 1.00) were also more common in the patient group but not significantly. Six patients had MC1 and MC2. Inter-rater reliability of MC classification was *κ* = 0.838.

### 2.2. Serum Biomarker with Modic Changes

Nine serum biomarkers related to disc and bone marrow connective tissue and basement membrane remodeling were measured (Table 2). In subjects with AnyMC, C4M was increased (+6.1%; *p* = 0.048; 25.6 ng/mL) and C3M (+5.4%; *p* = 0.091; 12.2 ng/mL) and PRO-C4 (+21.6%; *p* = 0.091; 250.6 ng/mL) tended to be increased compared to control subjects (24.1 ng/mL, 11.6 ng/mL, 206.0 ng/mL, respectively) (Appendix A
Table A1, Figure 1a). Other biomarkers showed no differences. In subjects with lumbar MC1, only PRO-C3 tended to be elevated (+9.2%; *p* = 0.092; 8.93 ng/mL vs. 8.18 ng/mL). In subjects with lumbar MC2, PRO-C4 was significantly up-regulated (+23.7%; *p* = 0.038; 254.9 ng/mL vs. 206.0 ng/mL), while C3M (+5.9%; *p* = 0.059; 12.3 ng/mL vs. 11.6 ng/mL) and C4M (+4.9%; *p* = 0.062; 25.3 ng/mL vs. 24.1 ng/mL) showed trends for elevation.

When the analysis was restricted to cLBP patients only, which reflects the clinical situation, C3M (+16.7%; *p* = 0.088; 12.2 ng/mL vs. 10.5 ng/mL), PRO-C3 (+15.0%; *p* = 0.071, 8.49 ng/mL vs. 7.38 ng/mL), C4M (+8.7%; *p* = 0.060; 25.6 ng/mL vs. 23.6 ng/mL), and PRO-C4 (+40.2%; *p* = 0.085; 250.6 ng/mL vs. 178.7 ng/mL) tended to be up-regulated in patients with AnyMC (Appendix A
Table A2, Figure 1b). In addition, PRO-C3 was significantly upregulated in patients with MC1 (+21.0%; *p* = 0.045; 8.93 ng/mL) and with MC2 (+17.3%; *p* = 0.033; 8.66 ng/mL) compared to patients without MC (7.38 ng/mL). The biomarker concentrations did not correlate with the total number of MCs within subjects (data not shown).

Effect sizes (Cohen’s *d*) were 0.51, 0.53, 0.52, and 0.61. Post-hoc power analysis indicated low power: 0.35, 0.37, 0.36, and 0.47 for PRO-C3, PRO-C4, C3M, and C4M, respectively.

### 2.3. Correlation Analysis

Correlation between biomarkers: C3M correlated positively with PRO-C4 (*r* = 0.71, *p* < 0.001) and C4M (*r* = 0.89, *p* < 0.001) (Appendix A
Table A3). C4M correlated also positively with PRO-C4 (*r* = 0.74, *p* < 0.001). C2M correlated negatively with huARGS (*r* = −0.48, *p* < 0.001). All other correlations between biomarkers were not significant.

Correlation of biomarkers with patient data: HuARGS correlated weakly with age (*r* = −0.38, *p* = 0.03). PRO-C4 correlated weakly with Oswestry Disability Index (ODI) in cLBP patients (*r* = 0.37, *p* = 0.02), but not anymore after adjustment for multiple testing (*p* = 0.18). All other correlations were not significant. Of the four biomarkers (PRO-C3, PRO-C4, C3M, C4M) that correlated with MC, PRO-C4, C3M, and C4M correlated with each other. PRO-C3 did not correlate with any other biomarker.

### 2.4. Receiver Operating Characteristics Analysis

Receiver operating characteristics were calculated for PRO-C3, PRO-C4, C3M, and C4M. These were the biomarkers that correlated either with AnyMC, MC1, or MC2, using all subjects or patients only. PRO-C3 had the largest AUC for MC1 and MC2, for all the subjects (MC1: 0.66, 95% CI: 0.48–0.84; MC2: 0.65, 95% CI: 0.47–0.83) and for patients only (MC1: 0.70, 95% CI: 0.50–0.89; MC2: 0.72, 95% CI: 0.53–0.90) (Table 3). When data were not stratified for MC type, PRO-C4 had the largest AUC when all subjects were considered (0.62, 95% CI: 0.59–0.64), and C3M had the largest AUC when patients only were considered (0.64, 95% CI: 0.46–0.82).

We next calculated ROC for AnyMC, MC1, and MC2 for combinations of PRO-C3 with PRO-C4, C3M, and C4M (Table 4). The model PRO-C3 + C4M had the highest AUC for AnyMC (all subjects and patient only) and MC1 (patient only). The model PRO-C3 + PRO-C4 had the highest AUC for MC2 (all subjects and patients only). The model PRO-C3 + C3M had the highest AUC for MC1 when considering all subjects.

Considering the clinical situation, where a test for the presence of MC in cLBP patients is desirable, the bivariable model PRO-C3 + C4M had the highest AUC 0.73 (95% CI: 0.56–0.90). Specificity was 78.9% (95% CI: 57.9–94.9) and sensitivity was 73.7% (95% CI: 47.4–94.7).

## 3. Discussion

We measured serum biomarkers of extracellular matrix turnover related to disc and bone marrow in subjects with and without MC and tested these biomarkers’ ability to predict MC. We found that biomarkers related to type III and IV collagen degradation (C3M and C4M, respectively) and formation (PRO-C3 and PRO-C4, respectively) correlated with the presence of MC in patients with cLBP (*p* = 0.060–0.088). PRO-C3 in combination with C4M had a moderate diagnostic value for MC in cLBP patients (AUC = 0.73, specificity = 78.9%, sensitivity = 73.7%).

These results agree with the pathomolecular mechanism of MC, in particular MC1, where bone marrow fibrosis has been described [2,10]. Type III and IV collagen are key bone marrow matrix constituents and known to be increased in bone marrow fibrosis [36]. From biomarker studies of other fibrotic diseases, including myelofibrosis, it is likely that an increase of PRO-C3 correlates with an increased type III collagen deposition. An increased C4M (and in some analysis PRO-C4) indicates a remodeling of tissue vascularization as type IV collagen is an important basement membrane constituent. This is in agreement with previous reports of decreased pro-angiogenic serum cytokines in MC patients, i.e., vascular endothelial growth factor (VEGF)-C, VEGF-D, tyrosine-protein kinase receptor 2 (Tie-2), VEGF receptor 1 (Flt-1), intercellular adhesion molecule 1 (ICAM-1), vascular cell adhesion molecule 1 (VCAM-1) [30]. Biomarkers related to disc degradation were not increased (C1M, C2M, huARGS), indicating that disc resorption is not a specific mechanistic target for MC, as previously reported [31].

Bone marrow fibrosis and accelerated disc degeneration are hallmarks of MC [2,3,10,11,33]. Extracellular matrix biomarkers related to these processes could be valuable in detecting MC at an early stage of the typically long treatment history of cLBP patients. In this study, we found that circulating biomarkers related to bone marrow fibrosis (PRO-C3, C3M, PRO-C4, C4M) but not to disc degeneration (C1M, C2M, huARGS) were increased in cLBP patients with MC. Our statistical models with the highest diagnostic accuracy for MC were combinations of PRO-C3 with PRO-C4, C3M or C4M, depending on the type of MC. Correlation analysis between the different biomarkers showed that PRO-C4, C3M, and C4M correlate significantly with each other, and hence, combining C3M, and C4M did not enhance diagnostic value. PRO-C3 did not correlate with PRO-C4, C3M, and C4M, and, therefore, the combination of PRO-C3 with any of the other three biomarkers increases the diagnostic value. The best model in this study for AnyMC in cLBP patients was PROC3 + C4M and reached a moderate diagnostic value with an AUC = 0.73. While this is not satisfactory for clinical usage, the correlation of fibrosis-related serum biomarkers with MC indicates that bone marrow fibrosis is an active process in MC. Marrow fibrosis in MC has been mainly associated with MC1 but not MC2 [2,10]. Interestingly, we found significantly increased serum concentrations of PRO-C4 and PRO-C3 in subjects with MC2. The model PRO-C3 + PRO-C4 had a classification accuracy for MC2 with an AUC = 0.75 and a sensitivity of 100%. This suggests that bone marrow fibrosis is also a relevant pathomechanism in MC2. Histopathological studies of MC2 bone marrow will have to corroborate the fibrotic mechanisms in MC2.

The biomarkers PRO-C3, PRO-C4, C3M, and C4M that correlate with MC have been identified as markers for other fibrotic conditions. In myelofibrosis, a myeloproliferative and fibrotic disease of the bone marrow, serum PRO-C3 and PRO-C4 have already been described as potential biomarkers that correlate with increased type III and IV collagen in the bone marrow [36,37,38,39,40]. PRO-C4 has been identified as a serum marker to predict the progression of systemic sclerosis [41]. PRO-C3 has been suggested as a biomarker for several pathologies related to tissue fibrosis such as lung injury, viral and non-viral hepatitis, systemic sclerosis, vascular remodeling, and kidney diseases [42]. The epitope recognized by the PRO-C3 assay is generated by cleaving intact procollagen type III. The cleaved *N*-terminal fragment is liberated into the systemic level and measured in the blood serum. Hence, PRO-C3 correlates stoichiometrically with type III collagen formation.

The type IV collagen neoepitope C4M is increased in idiopathic pulmonary fibrosis, chronic obstructive pulmonary disease, and liver fibrosis [43]. The epitope recognized by the C4M assay is exposed when matrix metalloproteinase-2 and -9 (MMP-2, MMP-9) cleave intact type IV collagen, a major component of the basement membranes, including basement membranes from vascular sinusoids in the bone marrow. The activity of the gelatinases MMP-2 and -9 is locally increased in fibrotic tissue and disrupts the basement membrane, and allows the infiltration of fibroblasts and macrophage. Similarly, the type III collagen neoepitope C3M is exposed when MMP-9 cleaves intact type III collagen. C3M is a biomarker for liver fibrosis [44]. The specificity of these biomarkers for local fibrotic pathomechanisms grants them a high test and content validity for fibrotic disorders. Our data suggest that turnover of type III and IV collagen is increased in patients with MC and that increased serum concentrations could relate to MC because these patients were not known to have other fibrotic disorders.

Serum cytokines related to inflammation, bone turnover, angiogenesis, and vascular injury have recently been investigated as MC biomarkers [21,30,31]. However, no correlations were found with local pathophysiological processes. Therefore, these cytokines only have limited test validity. Cytokines have an average half-life in the range of minutes to hours [45], which allows the human body to react with a fast and dynamic cytokine response. Consequently, serum cytokine levels are very dynamic. In contrast, collagens are by the nature of their function thermodynamically very stable. For example, collagens of human intervertebral discs have an average half-life of 95–215 years [46]. Although collagen fragments may have a reduced half-life compared to intact collagen, they are more stable biomarkers than cytokines and are less affected by the diurnal cycle, food intake, and other acute events. In MC, inflammation is indicated by the edema seen on fat-saturated T2-weighted and on T1-weighted MR images. Yet, there is no direct evidence of inflammation in MC bone marrow [10]. Discs adjacent to MC were reported to secrete higher levels of inflammatory cytokines [10,24,26,27]. However, degenerated discs contain very few cells (<1 Mio) [47], and a small increase of inflammatory cytokine secretion is not detected on a systemic level, with no positive correlations with presence and size of MC [21].

The clinical relevance of MC is controversial [48,49]. Variability in the identification, classification, and reporting of MC contribute to discrepancies between study findings [50]. Besides known methodological factors related to imaging, time-varying changes in MC pathophysiology may contribute to the inconsistent findings. For example, fibrosis and bone resorption/high bone turnover are both hallmarks of MC [2,10,23]. These processes may occur in distinct phases. Therefore, serum biomarkers for bone marrow fibrosis, i.e., for PRO-C3, PRO-C4, and C4M, may not only help to detect MC but also help to stratify patients into a “fibrotic phase” and an “inflammatory bone resorption phase”. This could potentially inform treatment decisions and monitoring.

This study had several limitations. It was a single-site study with 54 participants with only 10 cLBP patients with MC1 and 11 cLBP patients with MC2. Variation in all biomarker concentration was high (Cohen’s *d* < 0.61 for AnyMC vs. no MC). Therefore, posthoc power analysis resulted in low power (power < 0.47). Despite the low power, differences in type III and IV collagen turnover suggest bone marrow fibrosis as a relevant MC pathomechanism. However, these findings need to be validated in independent cohorts. Given the limited number, it is important to note that there was no difference in gender, age, and BMI between patients and the control group and between subjects with and without MC. Nevertheless, results should be interpreted carefully and not over-generalized. The small participant number also limited the construction of more powerful statistical models. For example, including available patient data (sex, age, BMI) in biomarker models may improve the diagnostic accuracy. In order to avoid overfitting, we limited construct complexity to two biomarkers. Patient data were not included because they did not correlate with the presence of lumbar MC in our sample. Despite the limited data, fibrosis-related biomarkers were consistently upregulated and allowed for careful interpretation.

The causal relationship of serum type III and IV collagen with the pathological mechanism of MC remains unknown. Low local inflammation, as well as altered biomechanical loads, may both be factors contributing to bone marrow fibrosis. Clarification of these relationships requires cross-sectional histological studies as well as longitudinal studies that investigate the correlation of these biomarkers with MC progression and clinical parameters.

Bone marrow lesions (BML) in osteoarthritic joints share many characteristics with MC1, including pain association, natural history, local inflammation, and degradation of the adjacent cartilage [47]. Therefore, assessing biomarkers for type III and IV collagen in patients with osteoarthritic BML will gauge their specificity for MC and indicate pathomechanistic similarities between MC and BML in osteoarthritic joints.

MC size, type, and number may also affect serum biomarker concentration. In this current study, a correlation analysis of the number of MC lesions with serum biomarker concentrations revealed no significance.

Six patients had both MC1 and MC2, and some MC were of mixed type. Subgroup analysis for MC1 and MC2 showed similar biomarker trends. Nevertheless, MC1 and MC2 present different stages of the pathomechanism and have different histological features [2,10,11]. Therefore, larger, longitudinal studies are required in order to define accurate biomarkers for MC1 and MC2.

## 4. Materials and Methods

The study was conducted in accordance with the Declaration of Helsinki and approved by the site’s Institutional Research Board (#13-12224; approved 21 August 2014) and the local Ethics Commission (#2019-00845; approved 16 May 2019). Informed consent was obtained from each subject.

### 4.1. Study Subjects

The prospective cross-sectional study included 16 volunteers (no LBP, VASback ≤ 1), and 38 cLBP patients. Chronic LBP was defined as having LBP for more than three consecutive months and VASback ≥ 4 or ODI ≥ 30%. All subjects were recruited from outpatient clinics and from electronic medical records, and were enrolled between January 2016 and July 2018. Exclusion criteria were applied based on past medical history and MRI and were pregnancy, diabetes, positive smoking status, cancer, spondylolisthesis, scoliosis, prior lumbar surgery, disc herniation, compression fracture, and active use of osteoporosis medication. Patients and volunteers were age and sex-matched. All subjects completed a 10-item ODI questionnaire and indicated the intensity of back and leg pain on a 10-cm visual analog scale (VASback and VASleg, respectively).

### 4.2. Blood Serum Analysis

Peripheral blood was collected into serum-separating tubes (BD Bioscience). After 30 min at room temperature, the tubes were spun for 10 min at 1300× *g* at room temperature. The serum was frozen at −80 °C until analysis. Hemolysis was scored on a scale 0–3. All samples passed hemolysis test. The serum biomarkers were analyzed using enzyme-linked immunosorbent assays (ELISA). Each ELISA plate contained five controls and samples were run in duplicates in a blinded way. A coefficient of variance on duplicates of 15% was allowed. Samples with CV > 15% were rerun. Inter-assay variations from the five control samples are given in Table 2. No detectable amounts were set to the sensitivity level of the assay.

### 4.3. Magnetic Resonance Imaging (MRI)

The MRI of the lumbar spine was done at the time of blood serum collection in patients and volunteers. MRI was performed on a Discovery MR 750 3-T scanner using an eight-channel phased-array spine coil (GE Healthcare). Clinical fast spin-echo images with T1 and T2 weighting were acquired in the sagittal orientation with a field-of-view of 26 cm and slice thickness of 4 mm. Fat-saturated T2-weighted images had an echo time (TE) = 60 ms, repetition time (TR) = 4877 ms, echo train length (ETL) = 24, readout-bandwidth (rBW) = ±50 kHz, and an in-plane resolution of 0.6 mm. T1-weighted images had TE = 15 ms, TR = 511 ms, ETL = 4, BW = ±50 kHz, and an in-plane resolution of 0.5 mm.

Lumbar MC were graded by C.O. and Z.M. Both are board certified in pain medicine and have 20 years (C.O.) and eight years (Z.M.) of experience in reading and interpreting spine MRIs for clinical and research purposes. T1-and fat-saturated T2-weighted images were used to classify MC types according to the standard MC definition, which is MC1 = hypointense/hyperintense on T1/T2, MC2 = hyperintense/iso-to-hypointense on T1/T2, and MC3 = hypointense/hypointense on T1/T2 [50]. The cranial and caudal part of each lumbar vertebra was rated separately. To gauge inter-rater reliability of MC grading, both raters scored a subset of 45 vertebrae.

### 4.4. Statistical Analysis

Statistical analysis was done using R v. 3.6.2. For all tests, the significance level was set to α = 0.05. For power calculations, the power.t.test function of the package pwr in R was used. This was an exploratory study using convenience samples from patients that were recruited for a radiological study. In total, 42 cLBP patients and 18 healthy volunteers were recruited. One patient had to be excluded due to an infectious disease, and five blood samples had a nonacceptable level of hemolysis (>1). With the remaining *n* = 19 per group (AnyMC, no MC) in the cLBP population, an effect size of Cohen’s *d* = 0.91 was required to reach a power of 0.8. No preliminary data of the measured biomarkers were available for cLBP patients to gauge the effect size. A previous study in cLBP patients measuring hsCRP revealed a Cohen’s *d* of 1.46–1.72 [29]. A similar effect size would require a group size of *n* = 6–8. Post-hoc power was calculated for AnyMC for PRO-C3, PRO-C4, C3M, and C4M. Kappa statistic was used to assess inter-rater reliability of MC classification. For demographic information, differences between the two groups were tested with two-sided *t*-tests or Fisher exact tests for numeric and dichotomous data, respectively. The concentration of biomarkers were compared between two groups with *t*-tests or Wilcoxon rank-sum test if the Anderson–Darling test indicated non-normal distributed data. The correlations between biomarkers and patient information (age, sex, BMI) and between different biomarkers were tested with Pearson’s product-moment and corrected for multiple testing with the Benjamin–Hochberg correction (false discovery rate). Receiver operating characteristics (ROC), including the Area Under the Curve (AUC), sensitivity, specificity, and accuracy were computed using the pROC package in R (v. 1.16.1) for each biomarker that correlated with the presence of any type of lumbar MC (AnyMC), MC1, or MC2. Logistic regression models were calculated for combinations of two biomarkers. No sub-group analysis for MC3 was made because only two patients had MC3.

## 5. Conclusions

Serum biomarkers related to type III and IV collagen formation and degradation correlated with the presence of lumbar MCs in patients with cLBP. A multivariable statistical model, including PRO-C3 and C4M reached moderate diagnostic value to detect MC (AUC = 0.73). Together, these data indicate that bone marrow fibrosis in MC may be an important pathophysiological process that could be targeted in future biomarker studies that include more patients and quantitative MC evaluation.

## Figures and Tables

**Figure 1 ijms-21-03791-f001:**
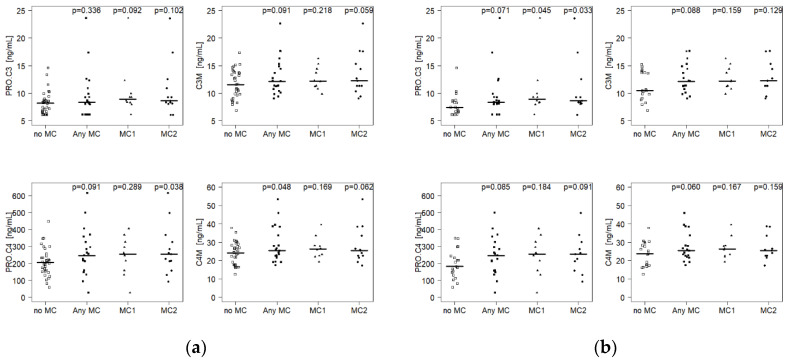
Biomarkers for turnover of type III and type IV collagen in (**a**) all subjects and (**b**) low back pain patients only. Horizontal lines indicate median value. *P*-values for comparison of Modic change (MC) groups (Any MC, MC1, MC2) versus no MC are given. AnyMC = any type of Modic change; MC1 = Modic type 1 change; MC2 = Modic type 2 change.

**Table 1 ijms-21-03791-t001:** Demographic information of low back pain (LBP) patients and control subjects.

Characteristics	LBP Patients (*n* = 38)			Control Subjects (*n* = 16)			LBP vs. Control
	AnyMC	No MC	*p*-Value	AnyMC	No MC	*p*-Value	*p*-Value
MC prevalence							
AnyMC	19 (50.0)	19 (50.0)		2 (12.5)	14 (87.5)		0.01 *
MC1	10 (26.3)	28 (73.7)	-	0 (0)	16 (100)	-	0.03 *
MC2	11 (28.9)	27 (71.1)	-	2 (12.5)	14 (87.5)	-	0.30
MC3	2 (5.3)	36 (94.7)	-	0 (0)	16 (100)	-	1.00
Demographics and clinical measures							
Male	8 (44.4)	12 (60.0)	0.33	1 (50.0)	7 (50.0)	1	0.21
Age	47.7 ± 12.1	47.9 ± 11.7	0.97	57.0 ± 9.9	42.3 ± 11.3	0.24	0.36
BMI	25.9 ± 6.0	25.8 ± 4.0	0.98	27.5 ± 2.6	23.1 ± 3.6	0.18	0.48
VASback	6.6 ± 1.8	6.5 ± 1.7	0.85	0 ± 0	0 ± 0	-	<0.001 ***
VASleg	3.6 ± 3.2	4.5 ± 3	0.41	0 ± 0	0 ± 0	-	<0.001 ***
ODI *	34 ± 14.2	31.9 ± 15.6	0.67	0 ± 0	0 ± 0	-	<0.001 ***

For continuous numerical variables (Age, BMI, ODI), mean values ± standard deviations are indicated. Group differences were tested with *t*-tests. For count data (Male), the count is indicated and the percentage is in the parentheses. Group differences were tested with Fisher exact tests. BMI: body mass index; VAS: visual analog score; MC: Modic change; AnyMC: any type of MC; MC1: MC type 1; MC2: MC type 2; MC3: MC type 3. Significant correlations are marked with asterisks: * *p* < 0.05, *** *p* < 0.001.

**Table 2 ijms-21-03791-t002:** Serum biomarkers.

Biomarker	Specification	Inter-Assay CV%	No. of Undetectable Samples
PRO-C1	Internal epitope in the *N*-terminal pro-peptide of type I collagen	<8.7%	2
PRO-C2	Released *N*-terminal pro-peptide of type II collagen	<8.0%	2
PRO-C3	Released *N*-terminal pro-peptide of type III collagen	<4.1%	11
PRO-C4	Internal epitope in the 7S domain of type IV collagen	<13.4%	0
C1M	Neo-epitope of MMP-2,9,13 mediated degradation of type I collagen	<12.5%	2
C2M	Neo-epitope of MMP-mediated degradation of type II collagen	<5.2%	0
C3M	Neo-epitope of MMP-9 mediated degradation of type III collagen	<4.5%	0
C4M	Neo-epitope of MMP-2,9,12 mediated degradation of type IV collagen alpha 1 chain	<15%	0
huARGS	*N*-terminal epitope of aggrecanase mediated degradation of aggrecan	<7.7%	0

CV: coefficient of variance.

**Table 3 ijms-21-03791-t003:** Area Under the Curve (AUC) of Receiver Operating Characteristics (ROC).

Biomarker	All Subjects			Patients Only		
	AnyMC	MC1	MC2	AnyMC	MC1	MC2
PRO-C3	0.58 (0.41, 0.74)	0.66 (0.48, 0.84)	0.65 (0.47, 0.83)	0.63 (0.44, 0.81)	0.70 (0.50, 0.89)	0.72 (0.53, 0.90)
PRO-C4	0.62 (0.46, 0.78)	0.59 (0.37, 0.82)	0.64 (0.45, 0.82)	0.63 (0.45, 0.82)	0.61 (0.38, 0.84)	0.63 (0.42, 0.83)
C3M	0.61 (0.45, 0.77)	0.60 (0.43, 0.78)	0.62 (0.43, 0.81)	0.64 (0.46, 0.82)	0.63 (0.44, 0.81)	0.63 (0.43, 0.84)
C4M	0.61 (0.45, 0.76)	0.59 (0.42, 0.77)	0.58 (0.40, 0.76)	0.62 (0.43, 0.80)	0.60 (0.41, 0.79)	0.57 (0.37, 0.77)

Only biomarkers with significant correlations or strong trends for correlation with Modic changes were evaluated. The mean value for Area under the curve (AUC) is indicated. 95% confidence interval in brackets. AnyMC = any type of Modic change; MC1 = Modic type 1 change; MC2 = Modic type 2 change.

**Table 4 ijms-21-03791-t004:** Receiver Operating Characteristics (ROC) of best bivariable biomarker models.

MC Type	Subjects	Model	AUC	Specificity (%)	Sensitivity (%)
AnyMC	all	PRO-C3 + C4M	0.67 (0.52, 0.83)	90.9 (51.5, 100)	52.4 (28.6, 85.7)
	patients	PRO-C3 + C4M	0.73 (0.56, 0.90)	78.9 (57.9, 94.9)	73.7 (47.4, 94.7)
MC1	all	PRO-C3 + C3M	0.69 (0.52, 0.86)	61.4 (43.2, 81.9)	90.0 (60.0, 100)
	patients	PRO-C3 + C4M	0.70 (0.51, 0.89)	67.9 (46.4, 85.7)	90.0 (60.0, 100)
MC2	all	PRO-C3 + PRO-C4	0.72 (0.55, 0.88)	75.6 (29.3, 100)	76.9 (38.5, 100)
	patients	PRO-C3 + PRO-C4	0.75 (0.60, 0.91)	59.3 (37.0, 100)	100 (54.5, 100)

Mean value indicated for Area under the curve (AUC), specificity, and sensitivity with a 95% confidence interval in brackets. Age, gender, and BMI were not included in the models because they did not significantly correlate with MC. AnyMC = any type of Modic change; MC1 = Modic type 1 change; MC2 = Modic type 2 change.

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
