# Peer review of "Serum Biomarkers for Connective Tissue and Basement Membrane Remodeling Are Associated with Vertebral Endplate Bone Marrow Lesions as Seen on MRI (Modic Changes)"

_ijms, 2020, doi:10.3390/ijms21113791_

Round 1

Reviewer 1 Report

. It is a very good study. Description of what collagen type III, IV can speific from MC.
. It seems that additional explanation is needed for the possibility of appearing in other lesions.
. In other words, it would be better to study the group of patients with bone marrow change due to other degenerative disease as a control group. For example, cases of degenerative knee problem.

Reviewer 2 Report

The authors investigated the serum biomarkers for connective tissue and basement membrane remodeling in association with vertebral endplate bone marrow lesions. However, there are some problems had to be detail clarified 

1. In introduction, author mention that they want to detect biomarkers reflecting connective tissue and basement membrane turnover for MC. Please, introduce these biomarkers.

  1. Extensive editing of English language is necessary. The article section had some problem.
  2. There was no formal hypothesis enunciated prior to the study, especially in mechanism.
  3. There was no sample size calculation done before the study and so it is not possible to say if the sample size was adequate for the conclusions drawn especially where statistical significance was not seen.
  4. It is necessary to report also all the new studies of the literature on this topic which are now missing. 
  5. The mechanism of pathophysiological links between MC and these markers is reported in some previous studies. We suggested authors to report current results in article.
  6. This article lack the important part, Western blot data, to confirm the mechanism. All article only mention the association without the causal relationship.
  7. The discipline, novelty and general significance of this study cannot reflect the quality of this journal.

Reviewer 3 Report

The manuscript is correct and relevant in its field. I just want to suggest a few comments regarding to the methodology and introduction:

I would suggest including Incidence ratios of specific pathology of vertebral endplate bone marrow lesions and its burden in different regions. This estimation would improve the context and impact of the study.

I would also recommend the mechanical multiscale function and some references showing the connection between biomechanics and biochemical factors in this problem is it is possible. 

The motivation of the number of samples involved in the study are omitted. Please provide a reason to consider this election. 

The exclusión criteria is not described in the methodology. As such as Normality tests. 

Round 2

Reviewer 2 Report

The authors have corrected many mistake and considered our suggestions.

We thanks author to provide detail answers to our questions.

Thanks!